# Development and Evaluation of Anti-Pollution Film-Forming Facial Spray Containing Coffee Cherry Pulp Extract

**DOI:** 10.3390/pharmaceutics17030360

**Published:** 2025-03-12

**Authors:** Weeraya Preedalikit, Chuda Chittasupho, Pimporn Leelapornpisid, Sheng Qi, Kanokwan Kiattisin

**Affiliations:** 1Department of Cosmetic Sciences, School of Pharmaceutical Sciences, University of Phayao, Phayao 56000, Thailand; weeraya.pr@up.ac.th; 2Department of Pharmaceutical Sciences, Faculty of Pharmacy, Chiang Mai University, Chiang Mai 50200, Thailand; chuda.c@cmu.ac.th (C.C.); pimporn.lee@cmu.ac.th (P.L.); 3School of Pharmacy, University of East Anglia, Norwich NR4 7TJ, UK; sheng.qi@uea.ac.uk

**Keywords:** coffee cherry pulp, film forming spray, anti-pollution, stability, penetration, safety testing, clinical trial

## Abstract

**Background/Objectives**: This study aimed to develop and evaluate an anti-pollution film-forming spray (FFS) containing coffee cherry pulp extract (FFS-CCS). The formulation was designed to create a protective skin barrier, improving skin health while defending against environmental pollutants. Its physical properties, dust resistance, stability, skin penetration, and clinical effectiveness were assessed to ensure optimal performance and safety. **Methods**: Various polymers and a ternary solvent system were used to enhance the stability and solubility of bioactive compounds from the coffee cherry pulp extract. The formulations were characterized based on appearance, film formation, viscosity, pH, spray uniformity, spray pattern, angle, film thickness, and particle adhesion. Stability testing was conducted under different storage conditions. Skin penetration was assessed using Franz diffusion cells with Strat-M^®^ membranes to simulate human skin. A single-blind, placebo-controlled trial with 42 participants was conducted over 60 days to evaluate the effects of FFS-CCS on skin hydration, tone, and wrinkle reduction. Clinical assessments were performed using a Corneometer, Mexameter, and Skin Visioscan. **Results**: The FFS1-CCS formulation, incorporating PVP K90 and a ternary solvent system, significantly improved the solubility, stability, and bioavailability of key bioactive compounds (chlorogenic acid, caffeine, and theophylline). Physical characterization confirmed uniform, transparent films with optimal viscosity and sprayability. Stability testing showed minimal degradation. Skin penetration and retention studies revealed enhanced retention of bioactive compounds with minimal systemic absorption. PVP K90, along with ethanol and propylene glycol, extended the compounds’ residence time on the skin, ensuring localized delivery. Clinically, FFS1-CCS significantly improved skin hydration, reduced roughness, lightened skin tone, and decreased erythema. **Conclusions**: The FFS1-CCS formulation utilizing PVP K90 significantly enhanced the stability, bioavailability, and skin retention of coffee cherry pulp extract, resulting in improved skin hydration, wrinkle reduction, and skin tone enhancement. These findings highlight the potential of coffee cherry pulp extract as a multifunctional, sustainable cosmeceutical ingredient, offering both anti-aging and environmental protection benefits, making it a promising solution for skincare applications.

## 1. Introduction

The skin, the largest organ of the human body, serves as a critical protective barrier against environmental aggressors, including pollutants [1]. Exposure to environmental pollution has been implicated in various adverse skin effects, such as accelerated aging, uneven skin pigmentation, and inflammatory skin diseases [2,3]. Anti-pollution products have thus been developed with the goal of not only protecting and cleansing the skin but also reinforcing the skin barrier, enhancing hydration, providing antioxidant protection, and mitigating inflammation. However, formulating a single product that effectively achieves all of these goals is challenging, often requiring the combination of multiple products [4]. One effective strategy is the creation of a physical barrier on the skin’s surface, which limits the direct contact of pollutants with the skin. Additionally, enhancing the penetration of active compounds, such as antioxidants and anti-inflammatory agents, can further strengthen the skin’s defense.

Film-forming spays (FFS) are innovative topical and transdermal formulations that combine active pharmaceutical ingredients (APIs) with film-forming excipients and volatile solvents, often presented as solutions or sprays. Upon application to the skin, the volatile solvent evaporates, leaving a residual film of excipients that acts as a reservoir for the API on the skin’s surface. This residual layer not only prolongs the contact time of the API but also ensures their controlled, gradual release, enhancing the efficacy of the API by promoting deeper skin penetration [5,6]. Films in situ have undergone substantial development, combining the advantages of films and hydrogels. FFS offers significant advantages over conventional formulations, including ease of application over large skin areas, precise unit dosing, and enhanced drug delivery. Their rapid drying and absorption properties minimize product transfer to clothing or other surfaces, making them particularly convenient for everyday use [7].

Coffee cherry pulp, a by-product of the *Coffea arabica* crop during coffee processing, is abundant in phenolic compounds, including chlorogenic acid (CGA), caffeine (CAF) and theophylline (THP) [8]. These bioactive compounds exhibit strong antioxidant, anti-inflammatory, and anti-aging effects [9,10,11]. The coffee fruit contains a broad spectrum of bioactive compounds, underscoring its value in anti-aging skincare formulations. Previous studies have demonstrated that coffee berry extracts, when encapsulated in nanoliposomes, exhibit enhanced stability, reduced toxicity, and improved anti-aging effects, including increased collagen synthesis, enhanced skin penetration, and improved skin elasticity and brightness [12]. McDaniel’s study suggested that topical formulations containing coffee berry extracts can significantly improve skin appearance, reduce wrinkle visibility, and enhance hydration while also modulating inflammatory pathways by decreasing MMP-1 and IL-1β levels, increasing collagen protein expression, and downregulating MMP gene expression [13]. Our previous research showed that coffee cherry pulp (CCS) extract, which is rich in CGA, CAF, and THP, significantly inhibits the production of pro-inflammatory cytokines, reactive oxygen species, and inflammatory enzymes in a dose-dependent manner following exposure to polycyclic aromatic hydrocarbons (PAHs), a major toxic component of particulate matter. The extract demonstrated comparable efficacy to its key constituents and showed no irritation in safety assessments, highlighting its potential as a natural cosmeceutical ingredient for mitigating skin inflammation induced by air pollutants [14]. Although the well-documented biological activities of coffee cherry extract, and cosmetics containing coffee cherry extract are commercially available, its incorporation into film-forming sprays for skin protection remains largely unexplored. This study aimed to develop an anti-pollution film-forming facial spray incorporating CCS extract. Comprehensive evaluations of the formulation’s physical characteristics, dust protection property, stability, and clinical efficacy were conducted to ensure the product’s performance and safety, providing a novel approach to enhance skin protection against environmental pollutants.

## 2. Materials and Methods

### 2.1. Materials

Polyvinylpyrrolidone K90 (Kollidon VA 64) and vinylpyrrolidone-vinyl acetate copolymer (Kollidon VA 64) were purchased from Fisher Scientific (Loughborough, UK). Polyacrylate crosspolymer-6 was obtained from Seppic (Paris, France). Propylene glycol (99%), ethanol, methylene blue, chlorogenic acid, caffeine, and theophylline were purchased from Sigma-Aldrich (St. Louis, MO, USA). Sodium lauryl sulfate (SLS) was purchased from Chanjao Longevity (Bangkok, Thailand).

### 2.2. Preparation of Coffee Cherry Pulp Extract

The coffee cherries were derived from *Coffea arabica* L. cherries harvested in December 2021 from Doi Chang, Chiang Rai Province (Figure 1a). The coffee cherry pulp, a by-product of coffee processing, was collected (Figure 1b), and further dry processed using hot air oven at 50 ± 2 °C to obtain dried pulp material (Figure 1c). The dried coffee cherry pulp was ground into a fine powder (Figure 1d) before undergoing extraction.

The extraction process followed the Soxhlet method, as detailed in our previous study to obtain coffee cherry pulp extract (CCS extract) [8]. Soxhlet extraction was chosen over ultrasound-assisted extraction (UAE) and maceration due to its continuous solvent reflux, which ensures efficient extraction and consistent recovery of bioactive compounds. While our previous study found no significant difference in % yield between Soxhlet and UAE, Soxhlet extraction yielded higher concentrations of chlorogenic acid, caffeine, and theophylline, along with superior antioxidant, anti-aging, and anti-inflammatory activities. Given the importance of bioactivity and compound stability, Soxhlet was determined to be the most suitable method for obtaining a standardized extract with optimal therapeutic potential.

Extraction was performed using 95% ethanol with a solid-to-liquid ratio of 1:5 g/mL, for 60 min. The resulting crude CCS extract (Figure 1e) was filtered and concentrated, then stored in an airtight container at a controlled temperature of 4–8 °C to preserve its bioactive properties for subsequent use.

### 2.3. Solubility Classification of CCS Extract

The solubility classification of the CCS extract was determined according to the British Pharmacopoeia guideline. The estimated lower solubility limit of the extract was defined based on the values outlined in Table 1. To assess the solubility, different ratios of water, ethanol, and propylene glycol (PG) were employed, as shown in Table 2.

Briefly, 10 mg of CCS extract was accurately weighed, and 10 µL of the solvent mixture was incrementally added until the extract was fully dissolved. The mixture was continuously stirred at room temperature for 24 h to ensure complete dissolution. Clear solutions were observed, and solubility values (mg/mL) were determined in accordance with the British Pharmacopoeia. In this system, PG was used as a plasticizer, while ethanol and water served as the primary solvents to enhance the solubility and dispersion stability of the CCS extract.

### 2.4. Preparation of Film-Forming Spray

To prepare FFS, the optimal solvent ratio was determined and used as a vehicle to disperse the selected polymers. Various polymers, including polyvinylpyrrolidone K90 (PVP K90), vinylpyrrolidone-vinyl acetate copolymer (PVP/VA 64), and polyacrylate crosspolymer-6 (ACP), were evaluated to determine their ability to form a film upon solvent evaporation and their suitability for spray application. The base formulations were prepared as follows: FFS1 contained 0.6% *w*/*w* PVP K90, and FFS2 contained 0.6% *w*/*w* PVP/VA 64, as shown in Table 3. The selection of film-forming polymer was based on solubility in the chosen solvent system and compatibility with the CCS extract. Each base formulation was subsequently characterized and subjected to stability testing compared with formulations that incorporated the CCS extract. For the preparation of CCS extract-loaded formulations, FFS1-CCS and FFS2-CCS, each containing 1% *w*/*w* CCS extract, were developed by considering several key factors: its maximum solubility in the selected solvent system, its effective concentration as demonstrated in anti-inflammatory cell-based studies, and its skin permeation and absorption characteristics. This concentration ensures both effective therapeutic action and optimal stability within the formulation, allowing the extract to maintain its efficacy while integrating well into the polymer matrix.

Briefly, PVP K90 and PVP/VA 64 were dispersed in deionized water and homogenized using a mechanical stirrer. ACP was then gradually added to the homogenized polymer solution. The CCS extract, pre-mixed with propylene glycol and ethanol, was slowly incorporated into the polymer solution. The entire mixture was stirred continuously for 24 h until a homogeneous solution was achieved [15]. This approach ensured formulation stability and suitability for application as a film-forming spray, allowing the selected polymers to facilitate uniform film formation on the skin. The incorporation of CCS extract enhanced the overall functionality of the FFS, contributing to the effectiveness of the formulation for their intended use.

### 2.5. Characterizations and Stability Tests of Film-Forming Spray

#### 2.5.1. Physical Appearance and Film Formation After Drying of Spray

Visual inspection was used to assess the clarity and color of the spray solution. The sprays were then applied from a distance of 7 cm onto a clean glass slide, left to dry for 1 min, and visually examined under a microscope at 20× magnification (Microscope Temperature Stage, Linkam Scientific Instruments, Surrey, UK) to assess their texture and appearance after being sprayed [16].

#### 2.5.2. Viscosity and pH

The pH of the FFS was measured using a calibrated pH meter (Seven Compact S220, Greifensee, Switzerland). The viscosity of the formulation at room temperature was measured using a rheometer (TA Instruments, New Castle, DE, USA). Viscosity measurements were conducted to determine the viscosity limit of the spray solution that could be effectively sprayed [17].

#### 2.5.3. Uniformity of Weight per Spray

The weight of each spray was measured to evaluate dose uniformity [18]. The container’s weight was initially recorded. Then, five consecutive doses of the formulation were sprayed, and the container was reweighed. The average weight per dose was calculated by dividing the difference between the initial and end final weights of the container by the number of doses sprayed [19].

#### 2.5.4. Spray Pattern and Spray Angle

A piece of solvent-sensitive paper was clipped to a board, and 0.1% methylene blue was dissolved in the formulation before being sprayed onto the filter paper [20]. The spray nozzle was positioned 15 cm away from the paper. The resulting two-dimensional spray patterns were measured to determine the area covered. The radius of the spray dots was measured from different angles, and the spray angle was calculated using the following equation:$$Spray\;angle\left(\theta \right)=\tan^{-1}(l⁄r)$$
where *l* is the distance of paper from the nozzle, and *r* is average radius of the circle.

#### 2.5.5. Theoretical Film Thickness

The film thickness was measured following the ASTM E0252-06R13 standard [21] using the specified equation [22]:$$Thickness\;(cm)=\frac{Mass\;(g)}{Area\;\left({cm}^{2}\right)\times Density\;(g/{cm}^{3})}$$
where *mass* (g) refers to the measured weight of the uniformly cut section of the dried film, while *area* (cm^2^) represents its surface coverage, and *density* (g/cm^3^) corresponds to the material density of the film.

To determine the density of the FFS solution, 1 mL of the liquid formulation was pipetted and weighed at room temperature. The density was then calculated using the following equation [23]:$$Density\;(g/{cm}^{3})=\frac{Mass\;(g)}{Volume\;\left(mL\right)}$$

#### 2.5.6. Particle Adhesion

To evaluate the efficacy of FFS-CCS in preventing particle adhesion, pig skin was used as a model to mimic human skin, modified from the method proposed by Kassakul et al. [24]. The tested skin was prepared from the back of 6-month-old pigs that were cut into 8 × 8 cm and were carefully prepared by removing the fat layer. Each test skin was divided into test and control areas. The tested skin was immersed in PBS (pH 7.4) at 32 °C for 30 min to simulate normal skin conditions. The spray formulations were then thoroughly applied to the test areas and allowed to dry for 15 min. Post−application, the treated skin was transferred to a fine dust spraying box, modified from the design proposed by Wacharalertvanich et al. [25], as illustrated in Figure 2. Charcoal black powder (Merck; average particle size < 10 µm) was used as the model pollutant, with 5 mg weighed to generate dust levels of at least 25 µg/m³, monitored using a PM 2.5 air detector. Following dust exposure, a rinsing procedure was performed by pouring 30 mL of water slowly and evenly over the test area, followed by the application of 75 µL of neutral liquid soap. The area was then dabbed dry with a paper towel [26]. The degree of persistent carbon particle adhesion after washing was analyzed using a Skin Visioscan VC20 (Courage and Khazaka Electronic GmbH, Cologne, Germany) and a Skin-Colorimeter^®^ CL 400 (Courage and Khazaka Electronic, GmbH, Cologne, Germany) according to the difference in light intensity between test and control samples. Values of light intensity range from 0 to 255; the closer the value is to 0, the darker the image. These data were compared with the control areas to determine the effectiveness of the FFS in protecting the skin from dust adhesion and facilitating rinse-off. Consequently, the optimal formulation was chosen for the stability study.

#### 2.5.7. Stability Study of Film-Forming Spray Containing CCS Extract

The stability of selected FFS-CCS was assessed using an accelerated stability testing method involving 8 cycles of heating (45 °C for 24 h) and cooling (4 °C for 24 h). Additionally, each formulation was stored for 3 months at 4 °C, room temperature (RT), and 45 °C. During this period, the physical appearance, viscosity, pH, and the levels of chlorogenic acid, caffeine, and theophylline were analyzed using high-performance liquid chromatography (HPLC) at 280 nm. The mobile phase consisted of acetonitrile and 1% *v*/*v* acetic acid (15:85) with a flow rate of 1 mL/min over 20 min [8].

### 2.6. Skin Penetration and Skin Retention Studies of Film-Forming Spray Containing CCS Extract

#### 2.6.1. Skin Penetration Determination

The skin penetration of the selected FFS-CCS was assessed using Franz diffusion cells (Logan Instruments, Somerset, NJ, USA) with modifications based on the method outlined by Quiñones et al. [27]. In brief, the receptor chamber was filled with 7.5 mL of PBS (pH 7), which was maintained at 37 ± 2 °C and continuously stirred at 300 rpm. Synthetic Strat-M^®^ membranes (300 μm, EMD Millipore, Burlington, MA, USA) were employed to simulate human skin for transdermal diffusion testing, as these membranes mimic the layered structure of human skin, including the epidermis, dermis, and subcutaneous tissue [28]. The membrane was placed onto the receptor chamber and closed with a donor chamber. Franz diffusion cells with a diffusion area of 3.14 cm^2^ were used to study in vitro skin penetration. To compare FFS-CCS with a CCS solution dissolved in 95% ethanol (used as the control), 1 mL of each formulation was added to the donor compartment. At specific time intervals (1, 2, 4, 6, 8, and 12 h), 1 mL aliquots were withdrawn from the receptor medium. Each withdrawn aliquot was immediately replaced with an equal volume of fresh PBS medium to maintain constant conditions. The concentrations of chlorogenic acid (CGA), caffeine (CAF), and theophylline (THP) that permeated through the membrane were analyzed using HPLC.

#### 2.6.2. Skin Retention Determination

FFS-CCS was investigated for skin retention after 12 h of sample application. Subsequently, the membranes were washed three times with PBS, cut into small pieces, and extracted with methanol using a sonication bath for 15 min. The concentrations of CGA, CAF, and THP retained in the membrane were then quantified using HPLC.

### 2.7. Clinical Study

#### 2.7.1. Study Design and Participants

Prior to starting the clinical study, the protocol was submitted and approved by the Human Ethics Committee of the Faculty of Pharmacy, Chiang Mai University (Approval No. 003/2024/F), in accordance with the Declaration of Helsinki guidelines. The study was designed as a single-blind, placebo-controlled, randomized trial, involving a total of 42 participants evenly divided into two groups of 21 each.

The sample size was determined to provide 80% power to detect a statistically significant difference, with a two-sided significance level of 5% and an effect size of 0.9 [29]. Participants were selected based on specific inclusion and exclusion criteria, with the primary inclusion criterion being healthy individuals over the age of 20 without any congenital skin diseases. Participants who experienced adverse reactions, such as erythema, itching, stinging, burning, or other skin abnormalities, and those who did not complete the study were withdrawn from the trial.

Assessments were conducted at baseline and after 30 and 60 days of product application. The study focused on evaluating skin irritation and changes in skin appearance, specifically measuring skin hydration, skin wrinkles, and skin tone, with comparisons to baseline measurements. The study design and overall process are illustrated in Figure 3.

#### 2.7.2. In Vivo Human Skin Patch Test

In this study, a single application closed-patch test was performed on 42 participants, following the guidelines of the European Society of Contact Dermatitis (ESCD) for diagnostic patch testing [30]. The study involved the application of 20 µL each of the test product, placebo, and both negative and positive controls onto standard Fin chambers. These chambers were then applied on the participants’ upper arms for 4 h. Bare skin served as the negative control, while 2% sodium lauryl sulfate was used as the positive control. Following the removal of the patches, observations for erythema and edema were made after 24, 48, and 72 h of patch remover. These observations were assessed using the Draize scoring system [31]. Based on these assessments, the Primary Irritation Index (PII) was calculated to evaluate the irritation potential of the test substances [32] using the equation below:PII=[(∑erythema grade)+(∑edema grade)]÷(4×N)
where ∑*erythema grade* is the sum of erythema scores recorded on the first day and after 24, 48, and 72 h, and ∑*edema grade* is the sum of edema scores recorded on the same time points. *N* represents the number of participants. PII values above 0.5 indicate potential skin irritation.

#### 2.7.3. Clinical Evaluation

The efficacy of the products was evaluated using a modified method based on Poomanee et al. [33]. Forty-two healthy participants, both male and female, aged over 20 years, were recruited for the clinical trial. Participants were randomly assigned to two groups of 21 subjects using a blocked randomization approach. One group received the selected FFS-CCS formulation, while the other received a placebo. The placebo used in this study was the FFS1 base formulation without the CCS extract, ensuring that it corresponded to the active formulation in all aspects except for the absence of the bioactive components. Participants were instructed to apply their assigned product evenly across their entire face twice daily for 60 days.

This study was conducted under controlled conditions, with the measurement room maintained at a temperature of 25 ± 1 °C and a relative humidity of 75 ± 1% [34]. The effectiveness of the products was assessed by evaluating skin hydration, skin tone reduction, and wrinkle improvement. These assessments were conducted using the Corneometer^®^ CM825 (Courage and Khazaka Electronic GmbH, Cologne, Germany) for skin hydration, the Mexameter^®^ (Courage and Khazaka Electronic GmbH, Cologne, Germany) for melanin and erythema index, and the Skin Visioscan VC20 (Courage and Khazaka Electronic GmbH, Cologne, Germany) for direct visualization and analysis of skin wrinkles. Measurements were taken at baseline and after 30 and 60 days of product application, focusing on two skin regions: the forehead and cheeks. The results were averaged across these regions to ensure consistency and reliability and were then compared to baseline values to evaluate the overall impact of the treatments.

### 2.8. Statistical Analysis

The data were reported as mean ± standard deviation and analyzed using analysis of variance (ANOVA) at the 95% confidence level (*p* < 0.05). Variance analysis was performed using SPSS statistical software. Additionally, statistical significance was determined using a paired *t*-test for comparisons between the data of two time points.

## 3. Results and Discussion

### 3.1. Solubility Study of CCS Extract

Solubility is a critical factor in the development of effective pharmaceutical formulations, as it directly impacts the ability of active pharmaceutic ingredients (APIs) to be absorbed through biological membranes and reach their target site of action [35]. To ensure both stability and efficacy, the polymers and the CCS extract must remain stable within the selected solvent system. The efficient dissolution of the CCS extract in the selected solvent system is essential for achieving optimal absorption and therapeutic efficacy.

This study investigated various ternary co-solvent systems comprising ethanol, water, and PG, with a focus on their potential application in spray formulations and their ability to dissolve coffee cherry pulp extract. The selection of these solvents was based on their compatibility with the main ingredients of the spray formulation and their effectiveness in solubilizing bioactive compounds from the coffee pulp, as reported in previous studies [36,37].

Among the solvent systems tested, the mixture of ethanol, water, and PG in a ratio of 40:50:10 provided optimal solubility for the CCS extract. The study demonstrated that 10 mg of CCS extract was completely soluble in 1 mL of this solvent mixture, corresponding to a solubility requirement of 100 parts of solvent per 1 part of solute. According to the British Pharmacopoeia classification, this solubility level categorizes the CCS extract as sparingly soluble.

Interestingly, increasing the ethanol content did not enhance solubility [38]. In contrast, the solubility of the extract increased with the addition of PG, highlighting the significant role of PG in improving the extract’s solubility, even in small quantities. PG, known for its effectiveness in dissolving hydrophobic substances, plays a crucial role in this formulation by enhancing both the solubility and stability of the active ingredients [39,40]. However, incorporating PG at concentrations exceeding 15% *w*/*w* could potentially lead to reduced solubility of the CCS extract. This result might be attributed to several factors, including an imbalance in solvent polarity, saturation limits, and altered solvation dynamics. PG acts as a bridge between the hydrophilic (water) and lipophilic (ethanol) components of the system. However, exceeding the optimal PG concentration may disrupt the delicate balance among the solvents, thereby negatively impacting the solubility of key bioactive compounds. Additionally, reducing the ethanol content to 35% diminishes its capacity to dissolve slightly lipophilic compounds, which could further compromise the overall solubility of the extract.

To achieve optimal solubility, stability, and film formation, the FFS-CCS formulation was developed using a balanced solvent system consisting of ethanol (40%), water (50%), and PG (10%). Ethanol was chosen as the primary organic solvent due to its ability to dissolve both hydrophobic and semi-hydrophilic compounds while also facilitating the rapid drying of the spray. Water served as the primary polar solvent, ensuring adequate solubilization of hydrophilic components, particularly chlorogenic acid, while maintaining biocompatibility and skin tolerance. PG functioned as both a plasticizer and humectant, improving film flexibility, reducing brittleness, and enhancing skin hydration as well as the penetration of active compounds. This balanced solvent system is particularly beneficial for optimizing the formulation’s bioavailability when applied topically. The ethanol, water, and PG system at the optimal ratio effectively dissolved the bioactive components of the CCS extract, a crucial factor for maximizing the formulation’s therapeutic potential and efficacy in topical applications.

The 1% *w*/*w* concentration of CCS extract in FFS-CCS was selected based on findings from our previous study, which evaluated its antioxidant activity, anti-inflammatory efficacy, and cytotoxicity. This concentration was determined using a 100-fold IC_50_ value from DPPH and lipid peroxidation inhibition assays to ensure a strong antioxidant effect [8]. In cell-based studies, CCS extract exhibited significant anti-inflammatory activity at its Effective Concentration (EC_100_). To maximize bioactivity while maintaining cell viability, 1% *w*/*w* was chosen as a 100-fold EC_100_ starting point [14]. Additionally, this concentration demonstrated optimal solubility and stability in the ethanol–water-PG solvent system, effectively preventing phase separation or precipitation while ensuring homogeneous dispersion within the polymer matrix.

### 3.2. Characterizations and Stability Test of Film-Forming Spray

#### 3.2.1. Characterizations of Physical Appearance and Spray Parameters

The physical appearance and film formation properties of film-forming sprays (FFS) are crucial parameters that significantly impact the efficacy of the final product. To assess these properties, the uniformity of films formed by different polymer-based formulations, with and without the CCS extract, was evaluated under 20× magnification after spray application and drying. The FFS base formulations were characterized as clear, transparent solutions, while the FFS-CCS formulations, containing the CCS extract, appeared as dark brown solutions. Upon application, both the FFS base and FFS-CCS produced thin, uniform coatings with a short drying time of approximately two minutes, ensuring convenience of use and user comfort.

However, distinct differences were observed between the two formulations. The film formed by PVP K90 (Figure 4a) showed some bubbles within the film, indicating higher viscosity (46.9 ± 5.2 mPa·s), which resulted in a slightly thicker film. In contrast, the film formed by PVP/VA 64 (Figure 4b) was thinner and had a smoother texture due to its lower viscosity (36.3 ± 5.1 mPa·s), leading to a less textured surface. This variation in film appearance can be attributed to differences in film thickness, with PVP K90 forming a slightly thicker (2.0 ± 0.0 µm) and more textured film compared to the smoother, thinner film of PVP/VA 64 (1.5 ± 0.0 µm). Notably, the incorporation of the CCS extract into the formulations did not affect the integrity of the films, as demonstrated in Figure 4c,d. This finding indicates that the CCS extract can be successfully added into these polymer matrices without compromising film quality, thereby maintaining both the uniformity and overall structural integrity.

The characterization of spray formulations revealed differences in viscosity, pH, spray angle, and spray weight, as summarized in Table 4. These parameters play a crucial role in determining the performance and usability of film-forming sprays, affecting their spreadability, application uniformity, and interaction with the skin. The pH values of the FFS formulations ranged from 4.5 to 5.5, aligning well with the natural pH range of human skin, which is typically between 4.1 and 5.8 [41]. This compatibility is essential for ensuring that the formulations do not disrupt the skin’s natural barrier, thereby minimizing the risk of irritation.

The viscosity of the formulations ranged from 35.9 mPa·s to 46.9 mPa·s, correlating closely with the ideal viscosity identified in previous studies, which is considered optimal for good sprayability and minimizes the risk of dripping during application [16]. FFS1, formulated with PVP K90, exhibited higher viscosity compared to FFS2, which contained PVP/VA 64. The higher molecular weight of PVP K90 contributed to the increased viscosity of FFS1 [42].

All formulations produced a spherical spray pattern, influenced by the specific nozzle design, and demonstrated proper spreading. The average spray angles ranged from 68.8 ± 1.8° to 71.1 ± 0.8°, reflecting the impact of viscosity on spray distribution. A spray angle below 85° is considered optimal, as it ensures effective coverage without excessive spreading, which could compromise the uniformity of the film [18,20]. The narrower spray angle of FFS1 (69.2 ± 0.5°), related to its higher viscosity, suggests a more concentrated spray, which may enhance film thickness. Conversely, the wider angle of FFS2 (71.1 ± 0.8°), resulting from its lower viscosity, allows for broader coverage and a more even distribution of the film. Notably, the incorporation of CCS extract into the base formulations did not affect sprayability or performance characteristics, ensuring balanced and uniform application while maintaining effective delivery.

The spray weight remained consistent across all formulations, ranging from 0.12 to 0.13 g. Additionally, the consistent spray weight ensures uniform application, which is essential for efficacy in practical use.

#### 3.2.2. Dust Protection Effect of Film-Forming Spray

The dust protection efficacy of the FFS formulations was evaluated by exposing pig skin to charcoal powder, with the intensity of dust deposition quantified using the L value from the Lab* color space. The L* value, which indicates lightness on a scale from 0 (black) to 100 (white), directly correlates with perceived brightness or intensity. Higher L* values reflect lighter areas, while lower values indicate darker areas [43]. This grayscale representation provides an objective measure of dust accumulation, where lower L* values correspond to higher dust accumulation, signifying the reduced protective efficacy of the FFS formulations. The results demonstrated that treatment with all FFS formulations significantly reduced the presence of dark particles on the skin compared to untreated controls. Standard camera images showed visibly fewer black particles in the treated areas, particularly with FFS1-CCS (Figure 5a), which displayed stronger protective effects than FFS2-CCS (Figure 5c). Visioscan images of FFS1-CCS (Figure 5b) and FFS2-CCS (Figure 5d) confirmed these findings, showing a marked reduction in dark spots and indicating a lighter skin tone in the treated groups, especially for FFS1-CCS. A quantitative analysis of the L* values further supported these observations, with FFS1 and FFS1-CCS showing significantly increased lightness compared to untreated skin and other formulations. These results suggest that FFS1 and FFS1-CCS provided a stronger protective effect than FFS2 and FFS2-CCS, indicating minimal dust accumulation. The superior performance of FFS1 highlights its enhanced dust protection, emphasizing the improved barrier properties of PVP K90 over PVP/VA 64. The enhanced protective efficacy of PVP K90 can be attributed to its film thickness, flexibility, and resistance to mechanical stress, which enhance its ability to resist dust adhesion even under prolonged exposure. PVP K90 was selected as the primary film-forming polymer for the FFS1-CCS formulation due to its excellent film-forming properties, biocompatibility, and ability to enhance bioactive retention on the skin. Its higher molecular weight and cohesive film properties enable PVP K90 to form a durable, continuous barrier on the skin, effectively minimizing gaps where dust particles could settle. Additionally, PVP K90 bridges the functionalities of gels and FFS transitioning from a liquid or semi-solid state to a cohesive, semi-solid film upon solvent evaporation, similar to a gel-based formulation [44,45]. In contrast, HPMC-based FFS formulations failed to form a stable film, instead producing rigid structures that lacked flexibility and adhesion, making them less effective than PVP-based formulations [22]. These attributes highlight the superior film-forming capabilities of PVP K90 in creating an effective protective layer against environmental pollutants.

Furthermore, the addition of CCS extract did not significantly impact dust protection efficacy, as the L* values remained comparable to those of FFS base formulations.

This finding indicates that CCS extract integrates well within the polymer matrix without disrupting the structural integrity of the film formed on the skin. 

In summary, while the FFS1 base formulation effectively functioned as a physical barrier against dust, the addition of CCS extract in the FFS1-CCS formulation offers significant therapeutic benefits for treating skin dysfunctions. The incorporation of CCS extract contributes bioactive properties that extend the formulation’s function beyond physical protection, providing a therapeutic effect for damaged or stressed skin. These combined results highlight the dual functionality of FFS1-CCS, which not only forms a protective barrier against pollutants such as dust but also promotes skin health through its anti-inflammatory and antioxidant properties. This dual benefit underscores the potential of CCS-enriched FFS formulations as comprehensive protective and therapeutic solutions, particularly for skin exposed to harsh environmental conditions.

#### 3.2.3. Stability Study of Film-Forming Spray Containing CCS Extract

Based on the characterization results, the most suitable formulation was selected by evaluating key properties, including pH, film-forming ability, and viscosity, which influenced the spray angle. Among the tested formulations, FFS1-CCS demonstrated the best overall performance and was therefore chosen for the stability study. The selected FFS1-CCS formulation contains 0.6% *w*/*w* PVP K90 and 1% *w*/*w* CCS extract, with a solvent system consisting of ethanol, water, and PG in a 40:50:10 ratio. The stability of FFS1-CCS was then assessed under various conditions to ensure its robustness and effectiveness. The formulation was subjected to eight cycles of heating and cooling, followed by storage at room temperature, 4 °C, and 45 °C for three months. The FFS1-CCS formulation demonstrated remarkable stability in terms of physical appearance and homogeneity throughout the study (Figure 6a). The pH remained consistent at 4.5, even after undergoing both accelerated and long-term stability testing, further indicating the long-term stability of the formulation. This stability can be attributed to PVP, which is a non-toxic, non-ionic, inert, temperature-resistant, and pH-stable polymer. Its biocompatibility and unique ability to interact with both hydrophilic and hydrophobic drugs enhance the formulation’s robustness, making it a highly reliable excipient for sustained stability [46,47].

Viscosity stability was also thoroughly evaluated, as shown in Figure 6b. On Day 0, the initial viscosity was recorded at 46.5 ± 7.6 mPa·s. After the heating and cooling cycles, a slight decrease in viscosity was observed, highlighting the formulation’s resilience to thermal stress. Over the 90-day storage period, the viscosity of the FFS1-CCS formulation remained relatively stable at both 4 °C and RT, with only minimal fluctuations. However, a more significant reduction in viscosity was observed at 45 °C, particularly after 60 days (45.0 ± 4.2 mPa·s) and 90 days (42.5 ± 1.0 mPa·s). Despite the viscosity reductions at 45 °C, the FFS1-CCS formulation maintained a stable physical appearance with no signs of CCS extract deposition or phase separation. The stability of the formulation can be primarily attributed to the PVP polymer used in its preparation. PVP was specifically selected for its role as an anti-nucleating agent and crystallization inhibitor, effectively preventing drug crystallization even after solvent evaporation [45]. Hui Cheng et al. found that PVP effective inhibits the nucleation of indomethacin, particularly at lower concentrations, compared to other polymers such as HPMC E5 and PVP/VA 64 [48]. Widely used in transdermal delivery systems, PVP plays a crucial role in maintaining formulation stability by acting as a powerful anti-nucleating agent.

In our previous study, we identified CGA, CAF, and THP as the active compounds in the CCS extract. The stability of these active compounds within the FFS1-CCS formulation was evaluated after 90 days under different storage conditions. After eight heating and cooling (H/C) cycles, CGA retention was 96.0%, indicating minimal degradation due to thermal stress. During storage at 4 °C and RT, CGA levels decreased gradually but remained above 90% even after 90 days, demonstrating the formulation’s effectiveness in preserving CGA under typical conditions (Figure 6c). At the elevated temperature of 45 °C, a more significant reduction in CGA was observed, with retention dropping to 88.5% by day 90. This result aligns with the findings of Mehaya et al., which indicate that chlorogenic acid gradually degrades during the roasting process [49]. These findings suggest that while the formulation protects CGA under normal storage conditions, prolonged exposure to high temperatures accelerates its degradation.

In addition, CAF demonstrated excellent stability across all storage conditions (Figure 6d). After H/C cycles, retention remained at 99%, and even after 90 days at 45 °C, 92% of CAF was retained. Under 4 °C and RT, only minor declines were observed, with retention remaining above 94% after 90 days. Many studies have shown that caffeine exhibits higher thermal resistance than chlorogenic acid and caffeic acids during roasting conditions [50,51]. The higher thermostability of caffeine could be attributed to its higher melting point (238 °C) compared to chlorogenic acid (207 °C) [52]. The high retention rates indicate that CAF remained highly stable within the FFS1-CCS formulation, ensuring its efficacy over the product’s shelf life. Barnes et al. conducted a study on the stability of a 10 mg/mL caffeine citrate oral formulation for neonatal use preserved with potassium sorbate. The study concluded that the formulation remained stable for up to one year at temperatures as high as 45 °C, with only minor changes observed in the pH and caffeine concentration [53]. Erickson et al. evaluated the stability of an enteral caffeine solution for preterm infants. Their study found that the formulation remained stable for six months at both 4 °C and 25 °C, with caffeine concentrations remaining within 5% of the target value. HPLC analysis confirmed formulation consistency across multiple research sites [54].

Due to their similar methylxanthine structures, with slight variations in methyl group arrangements, THP exhibited stability comparable to CAF. After six H/C cycles, 98% of THP remained, with retention slightly decreased to 97% at 45 °C by day 90 (Figure 6e). Under 4 °C and RT, THP retention remained above 98% after 90 days. The results demonstrate that the FFS1-CCS formulation effectively maintains the stability of CGA, CAF, and THP with high retention rates of the active compounds under standard storage conditions. This stability is likely attributed to the stabilizing effects of PVP, which prevents degradation and inhibits crystallization of the active compounds. Moreover, the high retention rates of the active compounds validate the formulation’s design, making it suitable for prolonged use with consistent therapeutic efficacy. However, the low temperature of 4 ± 2 °C was found to be suitable for maintaining a high content of CGA, ensuring the formulation remains effective across various conditions. The phytochemical stability was significantly higher when protected from heat, emphasizing the importance of proper storage to preserve efficacy.

### 3.3. Skin Penetration and Skin Retention Studies

The skin penetration and retention profiles of CGA, CAF, and THP from the FFS1-CCS formulation over 12 h are illustrated in Figure 7, with a focus on comparing results with the CCS solution. Notably, both CAF and THP permeated through the synthetic membrane, reaching the receptor chamber of the Franz diffusion cell from both the FFS1-CCS formulation and the CCS solution, as shown in Figure 7a. Specifically, CAF demonstrated higher penetration from the CCS solution with a steady increase over time, whereas the FFS1-CCS formulation resulted in consistently lower penetration levels. Interestingly, CGA exhibited delayed penetration, not being detected until the 12-h mark, indicating a slower release profile from both the CCS solution and the FFS1-CCS formulation. This penetration behavior aligns with the well-established 500 Dalton rule, which states that the stratum corneum is nearly impermeable to molecules with molecular weights exceeding 500 Da [55]. Moreover, the stratum corneum presents a higher barrier for hydrophilic molecules compared to lipophilic ones, making it more challenging for water-soluble compounds to penetrate the skin layers [56]. Although CAF and THP are hydrophilic, both of them are small molecules with molecular weights considerably below the 500 Da threshold. This characteristic enables them to penetrate systemic circulation easily. In contrast, the larger and more complex structure of CGA, with its hydrophilic nature, results in slower and more limited penetration into the stratum corneum, which is consistent with its delayed detection in the study. While skin penetration can lead to systemic absorption of bioactive compounds, this may be undesirable for cosmetic products intended for topical application due to the increased risk of systemic side effects and potential interactions with other medications [57]. To address this issue, PVP K90 was utilized as the primary polymer in the FFS formulation, serving as a controlled-release agent. Its role is to prolong the residence time of bioactive compounds on the skin, enabling gradual release, reducing systemic absorption, and enhancing targeted, localized effects. Previous studies have shown that increasing PVP content in films significantly improves skin permeability [58,59].

In contrast to skin penetration, the FFS1-CCS formulation demonstrated significantly higher skin retention than the CCS solution after 12 h, as shown in Figure 7b. The retention levels of CGA, CAF, and THP within the skin were significantly enhanced by the FFS1-CCS formulation, reaching 18.0 ± 1.1%, 25.7 ± 1.6%, and 20.6 ± 0.4%, respectively, compared to the CCS solution. The FFS1-CCS formulation is more effective in delivering these compounds topically, achieving higher concentrations in the skin while minimizing systemic exposure. The enhanced skin retention can be attributed to several factors associated with the excipients used in the FFS, particularly ethanol and PG, which improve the solubility of slightly more lipophilic compounds [60]. Previous studies have shown that ethanol and PG significantly enhance the skin permeability of low molecular weight compounds. PG, in particular, serves a dual role as a plasticizer and a penetration enhancer, significantly improving the delivery of active ingredients into the epidermis and enhancing transdermal drug delivery. By lowering the glass transition temperature (Tg) of the polymeric films formed in the formulation, PG increases film flexibility and permeability of the film, facilitating greater drug diffusion [61,62]. Therefore, the presence of these excipients plays a crucial role in enhancing the topical delivery and retention of bioactive compounds in the FFS1-CCS formulation, making it a superior choice for achieving localized effects with minimal systemic absorption. Additionally, the phytochemical profile of coffee cherry pulp indicates the presence of minor components, such as terpenes, which are known to act as a penetration enhancer by modifying the stratum corneum lipid structure. These components may further influence the delivery of bioactive compounds [63,64].

### 3.4. In Vivo Human Skin Patch Test

The skin irritation reaction of the FFS1-CCS was evaluated using the Finn chamber^®^ (SmartPractice, Phoenix, AZ, USA) after 4 h of application. Following the removal of Finn chamber^®^, slight erythema was observed at the 2% SLS application site in most volunteers (a PII range from 0.5 to 2.0). On the other hand, no erythema or edema was observed in areas where the FFS1-CCS and FFS1 base formulation was applied. The PII values of bare skin, 2% SLS, FFS1-CCS, and FFS1 were 0.02, 0.60, 0.02, and 0.03, respectively. These findings confirm the good safety profile of the formulation for human skin. These results are consistent with our previous study, which demonstrated that the CCS extract did not cause irritation in the hen’s egg chorioallantoic membrane assay, confirming its mild nature [14].

### 3.5. Clinical Evaluation

The effectiveness of FFS1-CCS was evaluated by comparing it to the base formulation (placebo). Skin topography images captured using the Skin Visiometer SV600 (Figure 8) demonstrate noticeable improvements in skin moisture and smoothness following the application of FFS1-CCS compared to the placebo. These findings were consistent with the skin moisture measurements obtained from the Corneometer^®^ CM 825 (Courage and Khazaka Electronic GmbH, Cologne, Germany). FFS1-CCS significantly increased skin hydration by 11.5 ± 1.3% on Day 30 and 18.5 ± 2.6% on Day 60. In comparison, the placebo only exhibited moderate improvements in hydration, with an increase of 2.6 ± 1.2% on Day 30 and 1.0 ± 1.2% on Day 60 (Figure 9a). The placebo formulation exhibited moderate improvements in skin hydration, which can be largely attributed to the high percentage of propylene glycol in the base formulation. Propylene glycol is a well-known humectant, meaning it has the ability to attract and retain moisture in the skin [62]. The occlusive characteristics of PVP are highly effective in forming a protective barrier on the skin, which reduces transepidermal water loss (TEWL) by preventing moisture evaporation [65,66]. The significant increase in skin moisture levels observed with FFS1-CCS can be attributed to the occlusive characteristics of PVP and the effect of propylene glycol as humectant. Additionally, the coffee pulp extract in FFS1-CCS is rich in cellulose, hemicellulose, and pectin, which are known for their hygroscopic properties [67,68,69,70]. These components help attract and retain moisture, enhancing the overall hydrating effect. The moisture level of the skin is a crucial factor that impacts the structure, appearance, and protective function of the skin [71].

In addition to enhancing skin hydration, FFS1-CCS also resulted in noticeable skin lightening (Figure 9b) and a reduction in erythema (Figure 9c) after 90 days of treatment compared to the placebo. The reduction in erythema observed with FFS1-CCS can be attributed to the anti-inflammatory potential of the CCS extract, particularly its protective effects against PAH-induced inflammation. PAHs are known to induce oxidative stress and inflammation by activating the NF-κB and AhR pathways, leading to increased levels of pro-inflammatory cytokines such as TNF-α and IL-6. This inflammatory cascade contributes to skin erythema and redness. The CCS extract, which contains chlorogenic acid, caffeine, and theophylline, has been demonstrated to effectively inhibit these inflammatory pathways, thereby reducing cytokine production and mitigating inflammation [14]. This effect likely contributed to the observed reduction in erythema, reflecting the CCS extract’s ability to alleviate pollutant-induced skin inflammation. Furthermore, the tyrosinase inhibition property of the CCS extract complements its anti-inflammatory actions, helping to reduce redness and improve overall skin tone, leading to enhanced skin health and appearance [8,14].

In terms of anti-aging benefits, FFS1-CCS was found to reduce skin wrinkles and roughness, as indicated by decreased grey levels beyond the threshold compared to the overall image roughness. The average skin roughness (R3) represents the arithmetic mean of the roughness across different segments. In contrast, the arithmetic average roughness (R5) is calculated by integrating the area formed by the middle line of wrinkle roughness and dividing it by its length, providing a measure of the mean roughness [72]. These parameters are measured in arbitrary units (A.U.), with lower values indicating reduced skin wrinkles. After 90 days of treatment, the FFS1-CCS significantly decreased R3, R5, and skin volume by 3.7 ± 1.4% (Figure 9d), 3.6 ± 1.1% (Figure 9e), and 2.8 ± 1.2% (Figure 9f), respectively, compared to the placebo. The minor improvements in skin roughness and smoothness observed with the placebo can be attributed to the combined effects of PVP K90 and propylene glycol. PVP K90 functions as a film-forming agent, creating a smooth layer that temporarily tightens the skin and reduces the appearance of fine lines, enhancing skin texture. Propylene glycol, a humectant, attracts and retains moisture, plumping the skin and making it appear smoother and less lined. Together, these ingredients provide a temporary anti-wrinkle effect by improving skin smoothness and reducing fine lines [37]. For longer-lasting results, these effects would benefit from the inclusion of more active anti-aging ingredients. The observed reductions in skin roughness and wrinkles confirm the consistency and reliability of the results, demonstrating the potential of FFS1-CCS for improving skin texture and smoothness.

Our previous research has shown that the CCS extract possesses the ability to inhibit hyaluronidase, collagenase, and elastase enzymes, along with protective effects against polycyclic aromatic hydrocarbon-induced oxidative stress and inflammation [8,14]. Although CGA, CAF, and THP were the primary focus of this study, other phytochemicals in the CCS extract, such as trigonelline, theobromine, protocatechuic acid, gallic acid, quercetin, catechin, epicatechin, procyanidin, and rutin, may also contribute to its anti-aging and protective effects [73,74]. Notably, the crude extract often exhibits greater antioxidant activity than individual isolated compounds, suggesting a synergistic effect among its components. This synergism enhances the overall antioxidant capacity, as different compounds may work through complementary mechanisms, leading to a stronger protective effect than their individual actions alone [8]. Among the key bioactive compounds, chlorogenic acid has demonstrated potent antioxidant and anti-inflammatory activities, while caffeine and theophylline further enhance anti-inflammatory effects, contributing to the formulation’s overall protective and therapeutic potential [14]. Saewan also reported the anti-aging benefits of coffee berry extract on both skin and hair cells [11]. These findings further support the anti-aging potential of FFS1-CCS, contributing to its effectiveness in improving skin texture and appearance. The combination of film-forming excipients with naturally occurring phytochemicals in the FFS1-CCS formulation enhances bioactive retention, making it an effective choice for achieving localized effects. Additionally, the polyphenols and flavonoids in CCS extract may reinforce its antioxidant activity, strengthening skin protection against oxidative stress and environmental pollutants.

## 4. Conclusions

The film-forming spray formulation containing CCS extract, utilizing PVP K90 as the main polymer in an optimized solvent system, effectively enhanced the stability and bioavailability of key bioactive components. Physical characterization confirmed the formation of uniform, stable films that provided effective protection against environmental pollutants. Clinical evaluations demonstrated that the formulation significantly improved skin hydration, reduced wrinkles, and lightened skin tone while causing no irritation, confirming its safety for use. These findings validate FFS1-CCS as an innovative and effective film-forming spray with substantial potential for cosmetic and cosmeceutical applications. The formulation offers enhanced anti-aging benefits and protective effects against environmental stressors. This novel approach utilizes film-forming technology to create a protective barrier on the skin while harnessing the antioxidant and anti-inflammatory properties of coffee cherry pulp extract, presenting a promising and underexplored avenue in skincare.

## Figures and Tables

**Figure 1 pharmaceutics-17-00360-f001:**
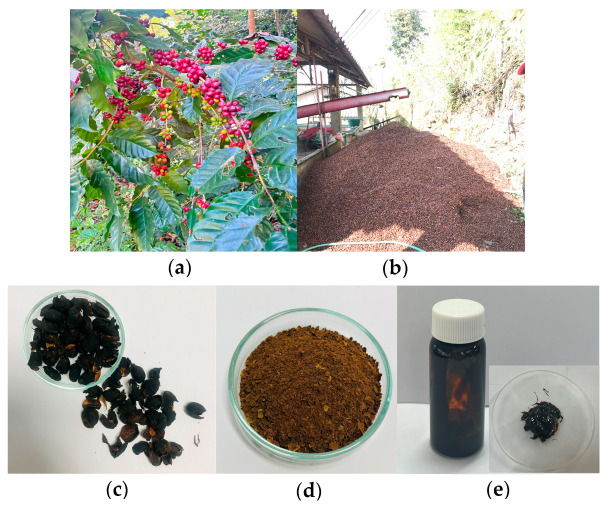
(**a**) Fresh coffee cherries from *Coffea arabica* L., (**b**) processing and collection of coffee cherry pulp as a by-product, (**c**) dried coffee cherry pulp before extraction, (**d**) ground coffee cherry pulp powder for extraction, (**e**) coffee cherry pulp extract (CCS extract) in semi-solid form.

**Figure 2 pharmaceutics-17-00360-f002:**
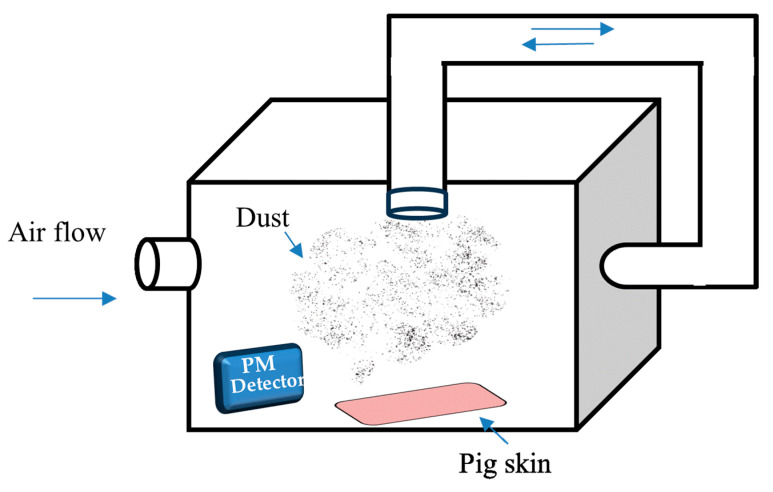
Schematic illustration of modified fine dust spraying box for dust protection performance test. The blue arrows indicate airflow direction within the system, ensuring uniform dust dispersion. The dust particles are introduced into the box and monitored using a PM detector, while pig skin samples are placed at the bottom to evaluate dust adhesion. The recirculation system helps maintain consistent particulate distribution during the test.

**Figure 3 pharmaceutics-17-00360-f003:**
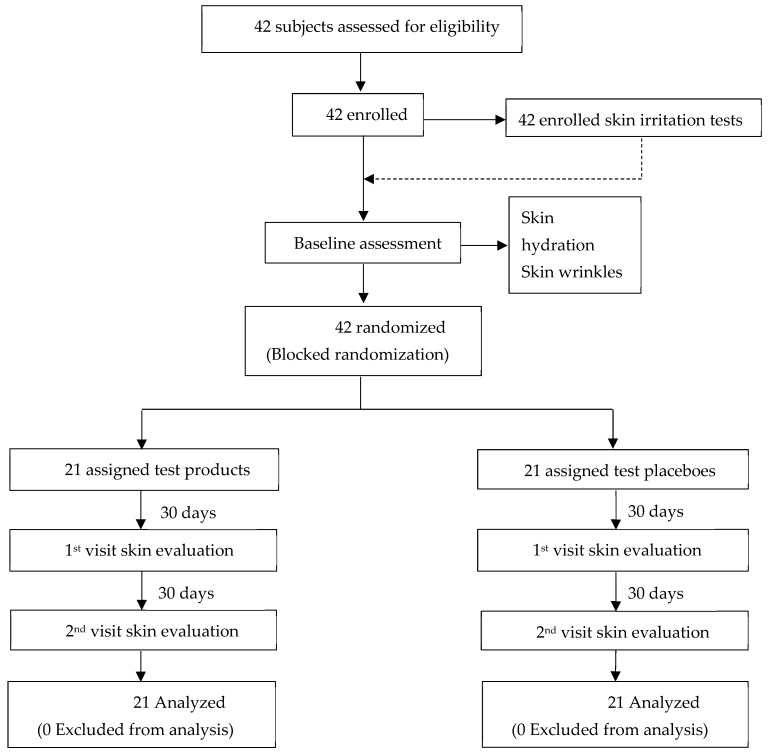
Flowchart illustrating the clinical trial process. A total of 42 subjects were assessed for eligibility, all of whom were enrolled and subsequently randomized into two groups using a blocked randomization approach. One group (*n* = 21) received the test product, while the other group (*n* = 21) was assigned placebo product. Skin evaluations were conducted at baseline and during the first and second follow-up visits. All 42 participants completed the study, and their data were included in the final analysis.

**Figure 4 pharmaceutics-17-00360-f004:**
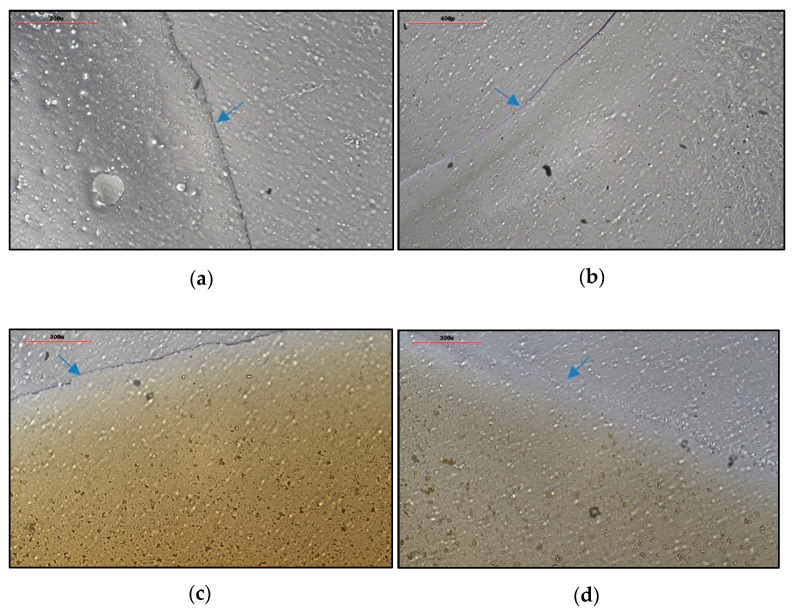
Appearance of (**a**) FFS1, (**b**) FFS2, (**c**) FFS1−CCS, and (**d**) FFS2−CCS after spraying under 20× magnification. The blue arrows indicate film boundaries in the sprayed formulations.

**Figure 5 pharmaceutics-17-00360-f005:**
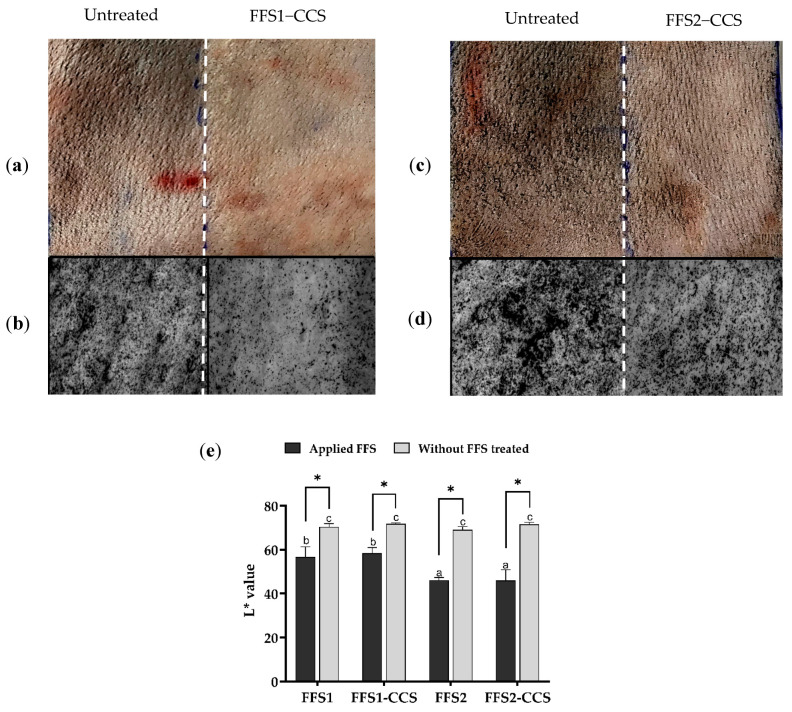
Comparison of FFS1, FFS1-CCS, FFS2, and FFS2-CCS formulations versus untreated control in dust protection efficacy using pig skin. (**a**) Standard camera image comparing untreated and FFS1-CCS-treated skin. (**b**) Visioscan image comparing untreated and FFS1-CCS-treated skin. (**c**) Standard camera image comparing untreated and FFS2-CCS-treated skin. (**d**) Visioscan image comparing untreated and FFS2-CCS-treated skin. (**e**) Quantitative analysis of skin lightness (L* value) comparing baseline, untreated, and treated groups. FFS1-CCS-treated skin showed less dust adhesion and higher L* values, indicating better dust protection and skin clarity than FFS2-CCS, confirming its superior barrier-forming ability. Statistical analysis (Tukey’s HSD test, *p* < 0.05) confirmed that groups sharing the same letters (a–c) were not significantly different. An asterisk (*) indicates a statistically significant improvement in dust protection compared to the untreated control. Data are expressed as mean ± standard deviation (S.D.) from three independent replicates (*n* = 3). (**b**,**d**) Visioscan images were captured using the Skin Visioscan VC20 with a scale bar representing 2 mm.

**Figure 6 pharmaceutics-17-00360-f006:**
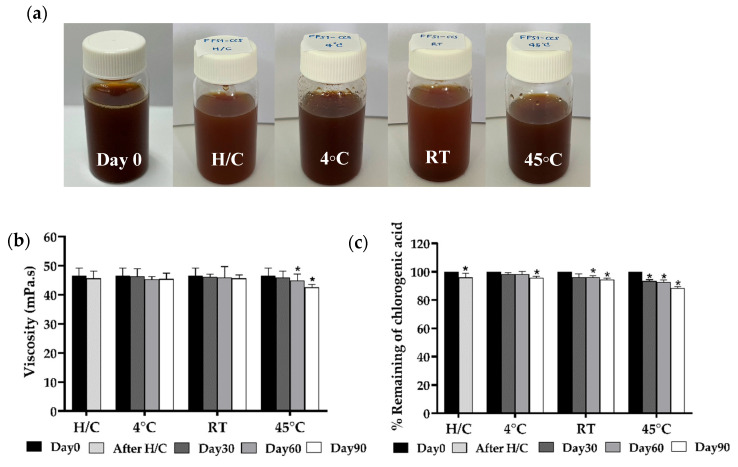
Physical stability profiles, including (**a**) physical appearance, (**b**) viscosity, (**c**) percentage of chlorogenic acid content, (**d**) percentage of caffeine content, and (**e**) percentages of theophylline content over 90-day storage under different conditions: heating and cooling cycle (H/C), 4 °C, room temperature (RT), and 45 °C. FFS1−CCS remained physically stable with no phase separation. Viscosity was consistent, except for a slight decrease at 45 °C. Chlorogenic acid degraded most at 45 °C, while caffeine and theophylline remained stable. Storage at 4 °C ensured optimal stability. An asterisk (*) indicates a significant difference (*p* < 0.05) compared to Day 0. Data are expressed as mean ± standard deviation (S.D.) from three independent replicates (*n* = 3).

**Figure 7 pharmaceutics-17-00360-f007:**
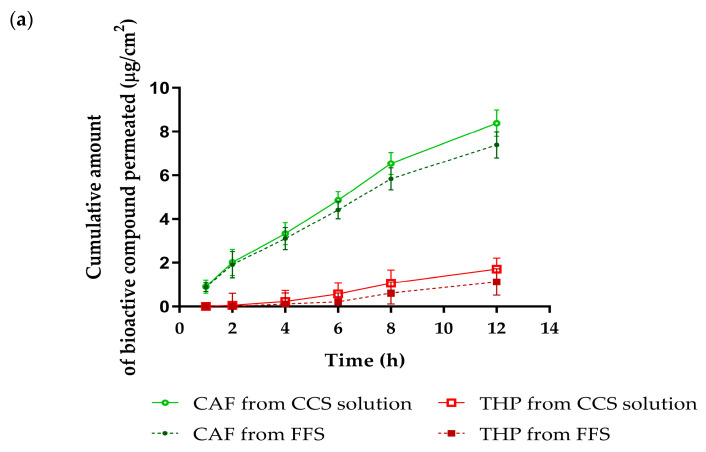
Skin penetration and retention profiles of FFS1-CCS compared to CCS solution. (**a**) Skin penetration study at various times within 12 h and (**b**) skin retention profile at 12 h of the FFS1-CCS formulation compared to the CCS extract solution. The FFS1−CCS formulation significantly improved skin retention of CGA, CAF, and THP, indicating localized delivery, reduced systemic absorption, and prolonged bioactive effects. Abbreviations: CGA, chlorogenic acid; CAF, caffeine; THP, theophylline. An asterisk (*) denotes a statistically significant difference (*p* < 0.05) compared to the CCS solution. Data are expressed as mean ± standard deviation (S.D.) from three independent replicates (*n* = 3).

**Figure 8 pharmaceutics-17-00360-f008:**
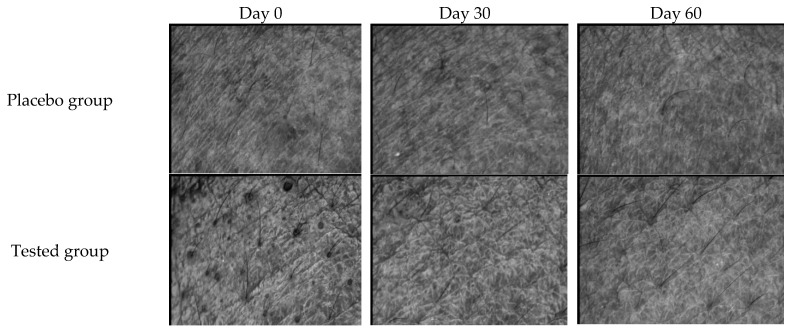
Skin Visioscan images illustrating the skin before (Day 0) and after application of the FFS base formulation (placebo group) and FFS1−CCS (tested group) for 30 and 60 days.

**Figure 9 pharmaceutics-17-00360-f009:**
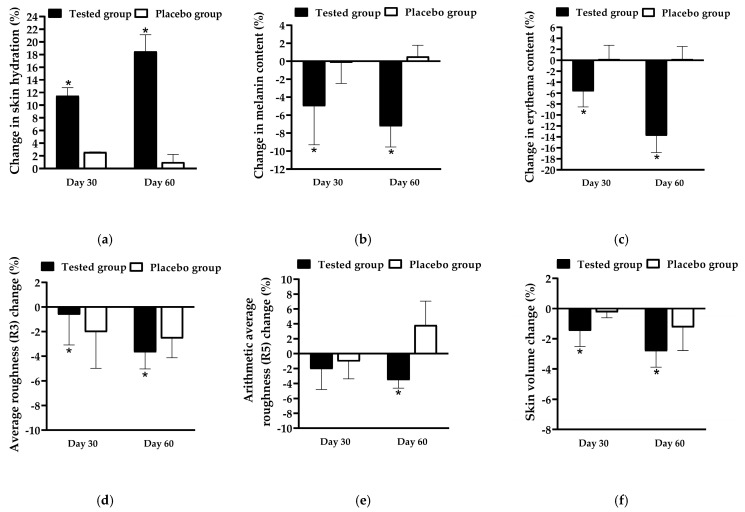
The change in (**a**) skin hydration, (**b**) melanin content, (**c**) erythema content, (**d**) average roughness (R3), (**e**) arithmetic average roughness (R5), and (**f**) skin volume before (Day 0) and after application of the FFS1−CCS (tested group) and FFS base formulation (placebo group) for 30 and 60 days. An asterisk (*) indicates a significant difference compared to that of placebo group at *p* < 0.05. Data are expressed as mean ± standard deviation (S.D.) based on 21 volunteers per group (*n* = 21 per group) and three replicates per participant (*n* = 3 per participant).

**Table 1 pharmaceutics-17-00360-t001:** Solubility definition.

Descriptive Term	Parts of Solvent Required Per Part of Solute	Descriptive Term
Very soluble	<1	>1000
Freely soluble	1 to 10	100–1000
Soluble	10 to 30	33–100
Sparingly soluble	30 to 100	10–33
Slightly soluble	100 to 1000	1–10

**Table 2 pharmaceutics-17-00360-t002:** CCS extract in ternary co-solvents.

Ethanol (% *w*/*w*)	Water (% *w*/*w*)	Propylene Glycol (% *w*/*w*)
50	50	0
45	50	5
40	50	10
35	50	15
50	50	0

**Table 3 pharmaceutics-17-00360-t003:** Compositions of the prepared film-forming spray base and film-forming spray containing coffee cherry pulp extract formulations.

Formular	Ingredient						
	PVP K90	PVP/VA 64	ACP	PG	Ethanol	Water	CCS Extract
FFS1	0.6	0	0.1	10	40	50	-
FFS2	0	0.6	0.1	10	40	50	-
FFS1-CCS	0.6	0	0.1	10	40	50	1
FFS2-CCS	0	0.6	0.1	10	40	50	1

PVP K90 = polyvinylpyrrolidone K90, PVP/VA 64 = polyvinylpyrrolidone/vinyl acetate copolymer 64, ACP = polyacrylate crosspolymer-6, and PG = propylene glycol.

**Table 4 pharmaceutics-17-00360-t004:** Viscosity, pH, spray angle, spray weight, and film thickness of FFS1, FFS2, FFS1−CCS, and FFS2-CCS.

Formular	pH	Viscosity (mPa·s)	Spray Angle (θ)	Spray Weight (g)	Film Thickness (µm)
FFS1	5.5	46.9 ± 5.2 ^b^	69.2 ± 0.5 ^a^	0.12 ± 0.0 ^a^	2.0 ± 0.1 ^b^
FFS2	5.3	36.3 ± 5.1 ^a^	71.1 ± 0.8 ^b^	0.13 ± 0.0 ^a^	1.5 ± 0.0 ^a^
FFS1-CCS	4.5	46.5 ± 7.6 ^b^	68.8 ± 1.8 ^a^	0.13 ± 0.0 ^a^	2.4 ± 0.0 ^b^
FFS2-CCS	4.6	35.9 ± 5.2 ^a^	70.9 ± 0.7 ^b^	0.12 ± 0.0 ^a^	1.6 ± 0.1 ^a^

The values are expressed as the mean ± standard deviation (S.D.) for three replicates (*n* = 3). Within the same column, different letters indicate statistically significant differences among formulations, with a significance level of *p* < 0.05, determined using Tukey’s Honest Significant Difference (HSD) test following a one-way ANOVA.

## Data Availability

Data are contained within this article.

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
