# Peer review of "Development and Evaluation of Anti-Pollution Film-Forming Facial Spray Containing Coffee Cherry Pulp Extract"

_pharmaceutics, 2025, doi:10.3390/pharmaceutics17030360_

Round 1

Reviewer 1 Report

Comments and Suggestions for Authors

The article submitted for review is very interesting and can be published after adding some important information:

1) The photo of the plant and its part from which the material for extraction is obtained should be presented in the manuscript.

2) The authors should also add the photo of the extract obtained and explain why the Soxhlet apparatus was chosen for extraction and not, for example, ultrasound-assisted extraction, which is currently very popular.

3) The manuscript should provide the full composition of the extract obtained, not only information about the content of chlorogenic acid, caffeine and theophylline as the main ingredients. Other ingredients of the extract can also affect the properties of the obtained films, e.g. anti-aging, protective effects, as well as the ability of active substances to penetrate different layers of the skin (there may be terpene compounds in the extract that support the penetration process) - the article should include additional considerations on this topic.

4) It would also be important to check the antibacterial activity of the obtained films, this would expand the possible applications of the protective films obtained by the authors.

5) The authors write about the antioxidant activity of the obtained films, have studies been conducted on the antioxidant activity of the produced formulations? What compounds in the extract can show such activity?

Author Response

Dear Reviewer of Pharmaceutics,

Title: Development and Evaluation of Anti-Pollution Film-Forming Facial Spray Containing Coffee Cherry Pulp Extract (Manuscript ID: pharmaceutics-3494499)

We greatly appreciate the valuable comments and suggestions provided by all reviewers. We have carefully read and responded each comment, responding to them point by point. The specific alterations in the manuscript in response to the reviewer comments are shown in yellow highlight for comments of reviewer 1 and green highlight for comments of reviewer 2.

We hope that all of the changes have adequately addressed the reviewers’ concerns, so with these improvements, we sincerely hope our work will be accepted for publication in the Pharmaceutics.

Sincerely yours,

Kanokwan Kiattisin

Reviewer 2 Report

Comments and Suggestions for Authors
  1. The authors should briefly discuss why PVP K90 was chosen over other common film-forming polymers like HPMC and Carbopol.
  2. The authors have reported using the optimized solvent system (ethanol, water, and propylene glycol having a ratio of 40:50:10). However, a further elaboration on the rationale behind choosing the ratio should be strengthened to justify the choice of the solvent system.
  3. The authors should provide a detailed discussion on the rationale for selecting 1% w/w CCS extract. Was the selection of the extract concentration based on pre-screening studies conducted by the authors, literature survey, or efficacy data?
  4. The authors should measure the thickness of the films.
  5. Even though the authors have mentioned the number of replicates in some cases, the information is not included across all sections. The authors should clearly state the number of replicates for each test.
  6. In Figures 4, 5, and 6, the authors should provide a summary of the results within the captions.
  7. In Figure 6, the graphs appear as wonky. The authors should improve the visual clarity of the graphs. The same is true for Figure 8.
Comments on the Quality of English Language

 The English could be improved to more clearly express the research.

Author Response

Dear Reviewer of Pharmaceutics,

Title: Development and Evaluation of Anti-Pollution Film-Forming Facial Spray Containing Coffee Cherry Pulp Extract (Manuscript ID: pharmaceutics-3494499)

We greatly appreciate the valuable comments and suggestions provided by all reviewers. We have carefully read and responded each comment, responding to them point by point. The specific alterations in the manuscript in response to the reviewer comments are shown in yellow highlight for comments of reviewer 1 and green highlight for comments of reviewer 2.

We hope that all of the changes have adequately addressed the reviewers’ concerns, so with these improvements, we sincerely hope our work will be accepted for publication in Pharmaceutics.

Sincerely yours,

Kanokwan Kiattisin

Round 2

Reviewer 1 Report

Comments and Suggestions for Authors

Most of my comments have been taken into account and the article can be accepted for publication in this form.

Reviewer 2 Report

Comments and Suggestions for Authors

Accept in present form